# 2-Hydroxypropyl-β-cyclodextrin Regulates the Epithelial to Mesenchymal Transition in Breast Cancer Cells by Modulating Cholesterol Homeostasis and Endoplasmic Reticulum Stress

**DOI:** 10.3390/metabo11080562

**Published:** 2021-08-23

**Authors:** Yiyang Zhao, Linkang He, Tian Wang, Lifang Zhu, Nianlong Yan

**Affiliations:** 1Department of Biochemistry and Molecular Biology, College of Basic Medical Science, Nanchang University, Nanchang 330006, China; ashelyzhaoyiyang@163.com (Y.Z.); l.he@se17.qmul.ac.uk (L.H.); avasweet1213@163.com (T.W.); juliefangl@163.com (L.Z.); 2Department of Biochemistry and Molecular Biology, Queen Mary College of Nanchang University, Nanchang 330006, China

**Keywords:** 2-hydroxypropyl-β-cyclodextrin, cholesterol metabolism, endoplasmic reticulum stress, TGF-β/Smad signaling pathway, epithelial-mesenchymal transition

## Abstract

Cholesterol metabolism affects endoplasmic reticulum (ER) stress and modulates epithelial-mesenchymal transition (EMT). Our previous study demonstrated that 2-hydroxypropyl-β-cyclodextrin (HP-β-CD) attenuated EMT by blocking the transforming growth factor (TGF)-β/Smad signaling pathway and activating ER stress in MDA-MB-231 cells. To further assess the detailed mechanisms between cholesterol metabolism, ER stress, and EMT, LXR-623 (an agonist of LXRα) and simvastatin were used to increase and decrease cholesterol efflux and synthesis, respectively. Here, we found that high HP-β-CD concentrations could locally increase cholesterol levels in the ER by decreasing LXRα expression and increasing Hydroxymethylglutaryl-Coenzyme A reductase (HMGCR) expression in MDA-MB-231 and BT-549 cells, which triggered ER stress and inhibited EMT. Meanwhile, tunicamycin-induced ER stress blocked the TGF-β/Smad signaling pathway. However, low HP-β-CD concentrations can decrease the level of membrane cholesterol, enhance the TGF-β receptor I levels in lipid rafts, which helped to activate TGF-β/Smad signaling pathway, inhibit ER stress and elevate EMT. Based on our findings, the use of high HP-β-CD concentration can lead to cholesterol accumulation in the ER, thereby inducing ER stress, which directly suppresses TGF-β pathway-induced EMT. However, HP-β-CD is proposed to deplete membrane cholesterol at low concentrations and concurrently inhibit ER stress and induce EMT by promoting the TGF-β signaling pathways.

## 1. Introduction

The endoplasmic reticulum (ER) has multiple functions, including protein synthesis, modification, and processing, and plays a crucial role in lipid metabolism. Thus, proper ER function is essential for cell fate [1,2]. Many endogenous and exogenous stimuli can disturb the function of the ER and cause ER stress, including metabolic disorders of lipids and proteins and ER calcium store depletion [3]. Lipids, including cholesterol, fatty acids, and even sphingolipids, participate in ER stress [2,3,4,5]. It is well documented that ER stress can affect cholesterol metabolism and homeostasis, which have been reported to be critical regulators of cholesterol synthesis, efflux, and uptake in macrophages [2,6,7]. However, cholesterol accumulation can also trigger ER stress. In fact, based on our previous research, ER stress activated by cholesterol accumulation causes endothelial dysfunction [8,9]. Furthermore, treatment with DHA (Docosahexaenoic Acid) strongly downregulates the cholesterol biosynthesis pathway and upregulates the ER stress response in breast cancer cells, which means lowering cholesterol biosynthesis can also trigger ER stress [10]. Evidently, perturbation of cholesterol homeostasis, including the accumulation of cholesterol in the ER and a strong decrease in cholesterol levels, may induce ER stress.

ER stress is involved in most cancers, leading to unfolded protein response that promotes epithelial-mesenchymal transition (EMT), survival, and drug resistance [10,11]. For instance, enhanced ER stress was demonstrated to induce EMT and subsequently increase cell migration in human non-small cell lung cancer cells and lens epithelial cells [12,13]. EMT is key to the malignant transformation of cancer cells. In this process, epithelial cells lose their differentiation characteristics and become mesenchymal cells, acquiring migratory and invasive abilities [14]. Breast cancer is the most frequently diagnosed cancer in women worldwide, and more than 90% of breast cancer deaths are related to tumor metastasis [15,16]. According to previous research, unlike in most other cancer cells, ER stress can block EMT in breast cancer cells [17]. For example, treating MCF-7 and MDA-MB-231 breast cancer cells with both pterostilbene and goniothalam was found to inhibit EMT by inducing ER stress [18,19]. Furthermore, we previously reported that in MDA-MB-231 cells, HP-β-CD suppressed the TGF-β pathway by decreasing the expression of TβRI, triggering ER stress and attenuating EMT [20]. However, the detailed mechanism remains to be elucidated.

β-Cyclodextrins are cyclic oligosaccharides that have a lipophilic cavity and a hydrophilic outer surface. As a result, β-cyclodextrin can be used as an ideal drug or gene carrier in the construction of targeted drug delivery systems. Recently, β-cyclodextrin has received much attention owing to its ability to chelate hydrophobic molecules, such as cholesterol and sphingolipids, and finally regulate the homeostasis of cholesterol and other lipids. For instance, β-cyclodextrin has been used to treat Niemann-Pick Disease Type C and to promote atherosclerosis regression by releasing trapped cholesterol from lysosomes and normalizing cholesterol levels in the liver [21,22,23]. With these properties, HP-β-CD is widely used to deplete cholesterol from cell membranes and lipid rafts, which can affect many signaling pathways [24,25]. Numerous studies have shown that HP-β-CD can inhibit the metastasis of triple-negative breast cancer cells by depleting cholesterol [25,26,27,28]. In addition, the effect of HP-β-CD on cellular cholesterol has been demonstrated to be closely related to their concentration and the cholesterol content of treated cells [29,30].

Cholesterol metabolism is widely known to be mainly associated with cholesterol efflux and biosynthesis, which can be modulated by various factors. The liver X receptors (LXRs), including LXRα and LXRβ, promote the transcription of a series of secretion-related genes, including adenosine triphosphate (ATP)-binding cassette transporter A1 (ABCA1) and ATP-binding cassette transporter G1 (ABCG1) to regulate various physiological activities, such as the metabolism and transport of cholesterol and fat formation; hydroxymethylglutaryl-coenzyme A reductase (HMGCR) is a rate-limiting enzyme of cholesterol synthesis; the low-density lipoprotein (LDL) receptor (LDLR) can elevate cholesterol uptake [31,32]. However, in this study, we mainly explored cholesterol efflux and biosynthesis related to ER stress following treatment with HP-β-CD and the relationship among cholesterol metabolism, ER stress, and EMT.

## 2. Results

### 2.1. HP-β-CD Decreases the Efflux of Cholesterol by Inhibiting the Expression of LXRα

Similar to our previous findings, in MDA-MB-231 cells, western blot analysis revealed that HP-β-CD concentrations lower than 6 mmol/L could increase the expression of ER stress markers GRP78 and CHOP, concurrently, it could inhibit EMT by increasing the expression of E-cadherin and decreasing the expression of vimentin. Therefore, we selected the concentration of 5 mmol/L HP-β-CD to inhibit EMT in the subsequent experiments (Figure 1A). After the treatment of HP-β-CD, we observed a decrease in the expression of LXRα, ABCA1, and ABCG1, suggesting a reduced cholesterol efflux.

To further investigate the effect of HP-β-CD on cellular cholesterol metabolism, we examined the cytotoxicity of LXR-623 to MDA-MB-231 cells by the MTT assay. LXR-623, an agonist of LXRα, was not found to be cytotoxic; instead, it was found to slightly promote cell viability (Figure 1C). Based on western blot analysis, LXR-623 triggered the expression of ABCA1 and ABCG1 to promote cholesterol outflow, and incubation with 3 μmol/L resulted in a 90.6% increase in ABCG1 expression. Therefore, this concentration was used in the subsequent experiments (Figure 1D). Interestingly, 3 μmol/L LXR-623 was found to induce ER stress and inhibit EMT, resulting in 45.3%, 36.3%, and 89.5% increases in the expression of Glucose-regulated protein 78 (GRP78), C/EBP homologous protein (CHOP), and E-cadherin, respectively. The expression of vimentin was decreased by 70.1% (Figure 1E). However, combination treatment with HP-β-CD and LXR-623 reversed HP-β-CD-induced ER stress and cholesterol efflux. This effect was accompanied by a decrease in GRP78 and CHOP, and an increase in ABCA1 and ABCG1 (Figure 1F,G). Thereafter, to analyze the distribution of cholesterol in MDA-MB-231 cells, Filipin and ER protein Calnexin co-immunofluorescence staining was performed. Treatment with HP-β-CD and LXR-623 alone was found to stimulate the accumulation of cholesterol in the ER. Furthermore, confocal microscopy analysis indicated that cholesterol distribution on the cell membrane was decreased. Interestingly, treatment with the combination of HP-β-CD and LXR-623 reversed these changes, and the cholesterol content in the ER recovered to the level observed in the control group (Figure 1H).

### 2.2. Disruption of Cholesterol Efflux Inhibits the Invasion and Migration of MDA-MB-231 Cells

To examine the association between ER stress and EMT, we carried out a western blot analysis. HP-β-CD or LXR-623 inhibited the transformation of MDA-MB-231 cells from an epithelial to a mesenchymal phenotype. Specifically, treatment with HP-β-CD and LXR-623 resulted in a 38.5% and 27.9% decrease in vimentin and a 41.9% and 45.6% increase in E-cadherin, respectively (Figure 2A). However, LXR-623 reversed the changes induced by HP-β-CD by upregulating vimentin expression and downregulating E-cadherin expression. The wound-healing assay confirmed that treatment with HP-β-CD and LXR-623 alone inhibited the migration ability of cells, which was significantly recovered following treatment with the combination of HP-β-CD and LXR-623 (Figure 2B). The transwell assay also demonstrated that HP-β-CD restrained the invasion ability of cells by 48.1%, which was reversed by LXR-623 (Figure 2C).

### 2.3. HP-β-CD Induces ER Stress by Inducing the Expression of HMGCR and Increasing the Synthesis of Cholesterol

To explore the impact of HP-β-CD on cholesterol synthesis in MDA-MB-231 cells, we examined its effect on the expression of HMGCR, which was upregulated as the concentration of HP-β-CD increased (Figure 3A). According to the MTT assay, the combination of 5 mmol/L HP-β-CD with different concentrations of simvastatin affect the viability of cells obviously when the concentration of simvastatin reached 1.6 μmol/L (Figure 3B). Considering the cytotoxicity, we selected 1.2 μmol/L simvastatin as the most appropriate concentration for subsequent experiments. To confirm the results caused by treatment with HP-β-CD and LXR-623 shown in Figure 1E,F, firstly, we combined 5 mmol/L and different concentrations of simvastatin to determine whether HP-β-CD-induced ER stress was recovered by simvastatin and we found that simvastatin rescued it in a dose-dependent manner (Figure 3C). Then, we established a specific group to determine the effect of HP-β-CD and simvastatin on ER stress and cell cholesterol metabolism, respectively (Figure 3D). Interestingly, similarly to previous results, we found that treatment with HP-β-CD and simvastatin alone induced ER stress; however, their combination was found to restore the expression of GRP78 and CHOP to normal levels. Likewise, Filipin staining and immunostaining for the ER protein, calnexin, revealed that treatment with HP-β-CD and simvastatin alone resulted in the accumulation of cholesterol in the ER, and a decrease in its distribution on the cell membrane. As expected, treatment with the combination of HP-β-CD and simvastatin reversed the above changes, and the cholesterol content returned to the level observed in the control group (Figure 3E). Such findings indicate that HP-β-CD-induced ER stress was caused by cholesterol accumulation in the ER, while LXR-623 and simvastatin alleviated ER stress by decreasing cholesterol accumulation in the ER.

### 2.4. Disruption of Cholesterol Synthesis Inhibits EMT in MDA-MB-231 Cells

Different concentrations of simvastatin inhibited HMGCR and reversed HP-β-CD-mediated suppression of EMT in a dose-dependent manner. Specifically, treatment with 1.2 mmol/L simvastatin resulted in a 57.4% decrease in E-cadherin and a 37.7% increase in vimentin (Figure 4A). The EMT of cells was inhibited after treatment with HP-β-CD and simvastatin alone; however, combination treatment with HP-β-CD and simvastatin restored EMT marker expression to levels near normal (Figure 4B). According to the wound healing and Transwell assays, simvastatin could recover the migration and invasive abilities of cells treated with HP-β-CD (Figure 4C,D). Therefore, LXR-623 and simvastatin counteracted the inhibitory effect of HP-β-CD on EMT by reducing ER stress.

### 2.5. Lower Concentrations of HP-β-CD Promote TGF-β/Smad Pathway-Induced EMT in MDA-MB-231 Cells

In contrast to the effect of higher concentrations of HP-β-CD (1 mmol/L) on EMT of MDA-MB-231 cells, low concentrations of HP-β-CD steadily induced EMT of cells (Figure 5A). Analysis of the expression of ER markers indicated that GRP78 and CHOP were downregulated by 62.9% and 70.3% at 0.8 mmol/L. These results indicate that HP-β-CD concentrations lower than 1 mmol/L have the opposite effect to HP-β-CD concentrations higher than 1 mmol/L (Figure 5B). To further elucidate the mechanism through which low concentrations of HP-β-CD promote EMT, we examined the expression of markers of the TGF-β/Smad pathway and found that p-Smad2 was upregulated in a dose-dependent manner, whereas Smad2 showed no meaningful changes (Figure 5C). Moreover, in contrast to high concentrations, after the treatment of low concentration of HP-β-CD, we observed a decrease in the expression of HMGCR, suggesting a reduced cholesterol synthesis. Concurrently, there was an increased expression of LXRα, implying a promoted cholesterol efflux (Figure 5D). By examining the distribution of cholesterol in cells by Filipin and ER marker staining using a confocal laser microscope, we found that 0.6 and 0.8 mmol/L HP-β-CD depleted cholesterol from the cell membrane, and instead of causing cholesterol accumulation in the ER, slightly lowered the amount of cholesterol in the ER (Figure 5E).

### 2.6. Effects of HP-β-CD on the Expression and Distribution of TβRI

TβRI is a receptor of TGF-β, which is distributed in both lipid rafts and no lipid raft parts of the membrane. The lipid raft TβRI is mainly involved in transmembrane signaling. Therefore, it is necessary to measure the expression and distribution of TβRI. As shown in Figure 6A, the expression of TβRI was upregulated after cells were treated with low concentrations of HP-β-CD (0.6 mmol/L and 0.8 mmol/L), and downregulated when treated with a high concentration of HP-β-CD (5 mmol/L). The co-staining of TβRI and lipid rafts (CTxB) of cells treated with HP-β-CD further indicated that low concentrations of HP-β-CD increased both TβRI expression and distribution in lipid rafts and the total area of lipid rafts in the membrane. High concentrations of HP-β-CD were found to reduce the distribution of TβRI in lipid rafts; however, the total area of lipid rafts in the membrane did not change (Figure 6B).

### 2.7. ER Stress Inhibits the TGF-β/Smad Pathway in MDA-MB-231 Cells

After MDA-MB-231 cells were treated with different concentrations ranging from 0–2 μmol/L of tunicamycin, an MTT assay was carried out. Based on the results, less than 2 μmol/L tunicamycin had no effect on cell viability. Thereafter, we opted to examine the expression of GRP78 and CHOP, which were upregulated in a dose-dependent manner following treatment with less than 2 μmol/L tunicamycin. Owing to this upregulation, 1 μmol/L tunicamycin was selected as the most appropriate concentration for use in the subsequent experiments (Figure 7A,B). To examine the relationship between ER stress and EMT, we treated cells with tunicamycin to induce ER stress and examined the expression of components of the TGF-β/Smad pathway and EMT. We found that the ratio of p-Smad2/Smad2 and the levels of vimentin and E-cadherin were decreased (Figure 7C–E). These results indicated that the TGF-β/Smad pathway and EMT were inhibited following the induction of ER stress by tunicamycin. Co-staining for TβRI and lipid rafts and confocal laser microscopy analysis revealed that the activation of ER stress decreased TβRI expression and distribution in lipid rafts. Interestingly, we found that ER stress increased the area of lipid rafts in the membrane (Figure 7F).

### 2.8. High Concentrations of HP-β-CD Disrupt Cholesterol Metabolism in BT-549 Breast Cancer Cells

To verify that HP-β-CD has the same effects on triple-negative breast cancer cells, BT-549 cells were treated with 0–10 mmol/L HP-β-CD. As a result, the expression levels of GRP78 and CHOP were found to decrease by HP-β-CD concentrations lower than 8 mmol/L and then inversely increased by HP-β-CD concentrations higher than 8 mmol/L up to 10 mmol/L (Figure 8A). Two representative concentrations (2 mmol/L and 8 mmol/L) were then selected to examine the levels of EMT and cholesterol metabolism marker proteins. The lower concentration of HP-β-CD (2 mmol/L) promoted EMT and induced a decrease in HMGCR and an increase in LXRα (Figure 8B,C). Further, a slight decrease in the cholesterol content in the membrane and ER of BT-549 cells was observed (Figure 8D). At the concentration of 8 mmol/L, HP-β-CD inhibited EMT proteins and led to the accumulation of cholesterol in the ER of cells (Figure 8B–D). These findings indicate that the triple-negative breast cancer cells had the same response when treated with HP-β-CD.

## 3. Discussion

Although some studies suggest that HP-β-CD could lower intracellular cholesterol [21,23], this study found that certain concentrations of HP-β-CD (5 mmol/L in MDA-MB-231 cells and 8 mmol/L in BT-549 cells) lead to the accumulation of cholesterol in the ER. Therefore, the impact of HP-β-CD on intracellular cholesterol was recognized to be closely related to its concentration, and the accumulation of cholesterol may be a negative feedback factor that increased the regional cholesterol in the ER [28,33].

The reason that led to cholesterol accumulation in the ER may associate with the LXRs which regulate cholesterol transport and outflow, namely cholesterol efflux, through promoting the transcription of a series of secretion-related genes, including ABCA1 and ABCG1 [32,34]. In this study, we discovered that HP-β-CD causes ER stress and decreases the expression of LXRα, ABCA1, and ABCG1. However, these changes were reversed by treatment with LXR-623. Thus, the downregulation of ABCA1 and ABCG1 was regulated by LXRα, which could promote the accumulation of cholesterol and ER stress. When LXRα was activated by LXR-623 (an agonist of LXRα), cholesterol efflux was promoted by ABCA1 and ABCG1, which lowered the accumulation of cholesterol and attenuated ER stress induced by HP-β-CD. These findings were also confirmed by Coisne et al., who reported that β-CDs decreased cholesterol efflux and the expression of ABCA1 and ABCG1 in smooth muscle cells and aortic endothelial cells [35]. Moreover, because the accumulation of cholesterol could trigger ER stress [20], HP-β-CD-induced ER stress may be related to the accumulation of cholesterol in the ER. For instance, Sozen et al. found that hypercholesterolemia plays a crucial role in triggering the IRE1/JNK branch of ER stress [36]. Our previous research also demonstrated that sphingomyelin synthase 2 (SMS2) can facilitate the accumulation of intracellular cholesterol, which contributes to the activation of ER stress and finally causes endothelial dysfunction [9]. In addition, we found that the expression of HMGCR was increased upon treatment with HP-β-CD, and simvastatin could abate cholesterol accumulation and ER stress. The increase in HMGCR promoted the biosynthesis and accumulation of cholesterol, which contributed to the triggering of ER stress. Simvastatin reversed these changes as it was found to inhibit the activity of HMGCR. In fact, other reports also indicated that HP-β-CD can increase cholesterol accumulation by upregulating cholesterol biosynthesis when the original cholesterol content of cells is normal [21]. Meanwhile, simvastatin attenuated ER stress by inhibiting the activity of HMGCR in endothelial cells [8]. Clearly, both LXRα and HMGCR modulate ER stress by affecting cholesterol homeostasis following treatment with HP-β-CD. By examining the total cholesterol content of MDA-MB-231 cells treated with HP-β-CD (data not shown), we found that there was no meaningful difference compared to the control group, which also suggested that cholesterol depletion by HP-β-CD was counteracted by the cholesterol homeostasis mechanism [35]. Together, the feedback effects of HP-β-CD could induce cholesterol accumulation in the ER and then trigger ER stress in MDA-MB-231 and BT-549 cells. Indeed, LDLR can also affect intracellular cholesterol [31]. In this study, we found that its expression was corresponding to the previous results in MDA-MB-231 and BT-549 cells, which verified the cholesterol feedback mechanism induced by HP-β-CD (Appendix A). However, its exact role requires further studies.

ER stress participates in many physiological and pathological processes, including EMT, and most studies indicate that ER stress can promote EMT [10,11,12,13,14]. However, we found that HP-β-CD suppressed EMT by triggering ER stress. Mechanistically, the effect of β-CDs on EMT of breast cancer cells is significantly related to their concentration [16,20]. For example, Zhao et al. found that when the concentration of M-β-CD is lower than 1 mM, the EMT of MDA-MB-231 breast cancer cells can be promoted; however, an opposite effect is observed when the concentration of M-β-CD is greater than 1 mM [16]. Wittkowski also showed that HP-β-CD can inhibit the migration of human MDA-MB-231 and MCF-7 breast cancer cells at concentrations of 1–4 mM [25]. Low concentrations of HP-β-CD can extract cholesterol from the cell membrane, especially from lipid rafts. At the same time, it can slightly decrease the levels of intracellular cholesterol; however, a high concentration of β-CD causes cholesterol accumulation in the ER [37]. Lipid rafts are mainly composed of sphingolipids and cholesterol. These rafts are crucial in physiological and biochemical processes and are recognized as “platforms” of signaling conduction [38]. When cholesterol is depleted from lipid rafts by low concentrations of HP-β-CD, the TGF-β/Smad pathway may be activated as the composition of lipid rafts can be affected and the levels of TGF-β receptor (TβRII/TβRI) in lipid rafts increase [18,39,40,41,42,43]. The TGF-β/Smad signaling pathway is one of the major signaling pathways that promote EMT; thus, low concentrations of HP-β-CD could induce EMT. The use of cholesterol-lowering reagents or other cholesterol-depleting reagents has been found to improve the TβRII/TβRI binding ratio and enhance TGF-β responsiveness, thereby promoting EMT in breast cancer [39]. This is also confirmed by the fact that the reduction of cholesterol levels in the mobile phase of the cell membrane increases the fluidity of the cell membrane and cell metastasis ability [44,45]. Moreover, our data showed that a low concentration of HP-β-CD inhibited ER stress, and previously, we found that treatment with TGF-β1 inhibited ER stress. These results suggest that a lower concentration of HP-β-CD could inhibit ER stress by activating the TGF-β/Smad pathway [20]. Taken together, our results suggest that low concentrations of HP-β-CD would mainly affect cholesterol in lipid rafts instead of intracellular cholesterol. However, treating MDA-MB-231 and BT-549 cells with high concentrations of HP-β-CD could lead to the accumulation of cholesterol in the ER, which could trigger ER stress and inhibit EMT. Further findings suggest that ER stress-mediated inhibition of EMT is also related to the TGF-β/Smad signaling pathway; this is because tunicamycin-induced ER stress was observed to attenuate the TGF-β/Smad signaling pathway, which is similar to the study of Huang et al., who found that tunicamycin, as an inducer of ER stress, can regulate multiple signaling pathways, such as those of AKT and MAPK [46]. Thus, we considered that low concentrations of HP-β-CD promoted EMT by activating the TGF-β/Smad signaling pathway, whereas, at high concentrations, HP-β-CD blocked EMT by inducing ER stress due to cholesterol accumulation in the ER and then inhibited the TGF-β/Smad signaling pathway.

Collectively, the findings of this research indicate that low concentrations of HP-β-CD depletes membrane cholesterol, concurrently inhibiting ER stress and inducing EMT by promoting the TGF-β signaling pathways. At higher concentrations, HP-β-CD could lead to the accumulation of cholesterol in the ER, inducing ER stress, which directly suppresses TGF-β pathway-induced EMT. Thus, we conclude that the TGF-β signaling pathway and ER stress are reciprocally regulated and that intracellular cholesterol homeostasis is a crucial factor. Additionally, because of the important role of cholesterol homeostasis in breast cancer, HP-β-CD may serve as a potential chemotherapeutic and adjuvant agent for cancer metastasis.

## 4. Materials and Methods

### 4.1. Cell Culture and Treatment

MDA-MB-231 and BT-549 cells were obtained from the Cell Bank of Type Culture Collection of the Chinese Academy of Sciences (Shanghai, China). The cells were maintained in a basal medium (RPMI-1640, Solarbio Science & Technology Co., Ltd., Beijing, China) containing 10% foetal bovine serum (FBS). To deplete cholesterol, cells were treated with HP-β-CD (BBI life Sciences, Shanghai, China) for 48 h. The regulation of cholesterol efflux and synthesis was carried out by culturing cells in an experimental medium containing LXR-623 (agonist of LXRα, cat. no. HY-10629; MedChemExpress, Monmouth Junction, NJ, USA), simvastatin, or DMSO (<0.1%) for 24 h. To induce ER stress, cells were incubated with tunicamycin (APExBIO technology, Houston, TX, USA, cat. no. B7417) for 24 h.

### 4.2. MTT Assay

Cells were cultured in 96-well plates in triplicate, and the dose-dependent effects of LXR-623, tunicamycin, and the combination of HP-β-CD and simvastatin on cell viability were determined. Following incubation, 20 μL of colorimetric 3-(4,5-dimethyl-thiazol-2-yl)-2,5-diphenyl-tetrazolium bromide (MTT) (5 mg/mL; cat. no. M8180; Solarbio Science & Technology Co., Ltd., Beijing, China) was added to each well, and after 4 h, the medium containing MTT was removed. Thereafter, 200 μL DMSO was added to each well to dissolve the formazan crystals. The OD490 was measured using a microplate reader (Thermo Fisher Scientific, Inc., Waltham, MA, USA).

### 4.3. Transwell Assay

After 8 h of starvation, trypsinisation, and counting, 2 × 105 cells in 200 μL serum free-medium were seeded into each well of the 24-well Transwell filters (24 wells; cat. no. 353097; Corning Life Sciences, Corning, NY, USA) covered with Matrigel (cat. no. 356234; Corning Inc., Corning, NY, USA). The lower chamber was filled with a complete medium containing 10% FBS. Cells were cultured for 48 h, fixed, and then rinsed twice with PBS at room temperature (25 °C). Finally, the cells were stained with 0.1% crystal violet (Solarbio Science & Technology Co., Ltd., Beijing, China). A microscope (Olympus iX71; Olympus corporation; magnification, ×40) was used to capture images and to count the number of invading cells.

### 4.4. Would Healing Closure

Cells were grown to 100% confluence and then starved with the experimental medium (1% FBS) in a 12-well plate for 6 h. Subsequently, the cells were cultured for 48 h, and a uniform straight wound line was created with a 200-μL tip in the middle of the well. At that time point (0 h), images of nine migration positions in each well were captured. After 48 h, images of the wounded cultures in the same positions were also captured to evaluate their migration ability (Olympus iX71; Olympus corporation; magnification, ×40).

### 4.5. Western Blot Analysis

The total cellular protein was extracted using RIPA buffer (cat. no. WLA019b; Wanleibio co., Ltd., Shenyang, China). The clear lysates (approximately 60 μg protein) were resolved on 10–12% SDS-PAGE gels and transferred onto a Polyvinylidene Fluoride (PVDF) membrane. The membranes were blocked in 5% non-fat dried milk or 5% bovine serum albumin (Solarbio Science & Technology co., Ltd., Beijing, China) for 1 h at room temperature and then incubated with primary antibodies overnight at 4 °C. After washing, the membranes were incubated with HRP-conjugated secondary antibodies for 1 h at room temperature. The antibodies were as follow: GRP78 (BIP/GRP78; cat. no. 66574-1-ig; 1:4,500; Monoclonal; ProteinTech Group, Inc., Wuhan, China), CHOP (DDIT3; cat. no. AP13063C; Polyclonal; 1:1000; Abgent, Ltd., Suzhou, China), ABCA1 (cat. no: BA154-2; Boster Biological Technology, Ltd., Wuhan, China), ABCG1 (cat. no. AP13063C; Polyclonal; 1:1000; Abgent, Ltd., Suzhou, China), LXRα (cat. no. 14351-1-AP; Polyclonal; 1:1,500; ProteinTech Group, Inc., Wuhan, China), HMGCR (cat. no. D154106; Polyclonal; 1:1,000; BBI life Sciences, Shanghai, China), E-Cadherin (cat. no. 20874-1-aP; Polyclonal; 1:4,000; ProteinTech Group, Inc., Wuhan, China), Vimentin (cat. no. 60330-1-ig; Monoclonal; 1:4,000; ProteinTech Group, Inc., Wuhan, China), TβRI antibody (cat. no. AF5347; 1:2,000; Polyclonal; Affinity Biosciences; ), LDLR (cat. no. 10785-1-AP; Polyclonal; 1:2,000; ProteinTech Group, Inc., Wuhan, China), phosphorylated-Smad2 antibody (cat. no. AF3450; 1:2,000; Polyclonal; Affinity Biosciences, Jiangsu, China), Smad2 antibody (cat. no. WL02286; 1:2,000; Polyclonal; Wanleibio co., Ltd., Shenyang, China), GAPDH antibody (cat. no. HRP-60004; 1:12,000; Monoclonal; ProteinTech Group, Inc., Wuhan, China), HRP-conjugated anti-mouse secondary antibody (cat. no. Sa00001-1; 1:12,000 ProteinTech Group, Inc., Wuhan, China) and anti-rabbit secondary antibody (cat. no. Sa00001-2; 1:10,000; ProteinTech Group, Inc., Wuhan, China). Finally, the membranes were incubated with ECL detection reagent (cat. no. WLA0006c, Wanleibio Co., Ltd., Shenyang, China), and the proteins on the membranes were visualized using a chemiluminescence detection system (Image lab version 5.1; Bio-rad laboratories, Inc., Hercules, CA, USA). Each western blot assay was repeated at least thrice.

### 4.6. Filipin Staining and Immunofluorescence

Filipin (cat. no. FF8614; Hefei Bomei Biotechnology Co., Ltd., Hefei, China) was dissolved in DMSO and then diluted to 50 µg/mL with PBS. Sterile coverslips (Cat. No. WHB-24-CS; WHB Scientific, Shanghai, China) were placed in a 24-well plate, and equal amounts of cells were seeded on them. After 48 h, cells were fixed at room temperature, washed three times with PBS, and incubated with glycine (1.5 mg/mL) for 10 min. Thereafter, cells were permeabilized with 0.1% Triton X-100 in PBS for 10 min and blocked with 1% BSA in PBS solution for 30 min at room temperature. The coverslips were then incubated with anti-calnexin primary antibody (cat. no. 10427-2-AP; Polyclonal; 1:500 dissolved in 1% BSA; ProteinTech Group, Inc., Wuhan, China) overnight at 4 °C followed by incubation with rhodamine-conjugated goat anti-rabbit antibody (cat. no. SA-00007-2; 1:50; ProteinTech Group, Inc., Wuhan, China) for 1 h at room temperature. Finally, 200 μL of Filipin working solution was added to each well and the cells were stained for 2 h at room temperature. The stained cells were observed under a confocal microscope (Olympus FV3000; Olympus corporation; magnification, ×400). The quantification of cholesterol accumulation in the ER was determined by ImageJ or Fiji analysis software. It was used to adjust the signal intensity thresholding and distinguish the region of interest (ROI). Then fluorescence co-localization (Mander’s correlation coefficient, MCC) was determined by evaluating fields acquired blindly in each group and by applying the same selection strategy [47]. The MCC obtained in each group was compared with the control group.

### 4.7. Immunofluorescence and Confocal Microscopy

Sterile coverslips (cat. no. WHB-24-CS; WHB Scientific, Shanghai, China) were placed in a 24-well plate, and equal amounts of cells were seeded on them. After 48 h, the cells on coverslips were fixed and blocked with 1% BSA/PBS solution for 30 min at room temperature. Thereafter, the coverslips were incubated with anti-TβRI primary antibody overnight at 4 °C followed by rhodamine-conjugated goat anti-rabbit antibody for 1 h at room temperature. Subsequently, Alexa Fluor-498-labelled CTxB (cat. no. C-34775; diluted at 10 μg/mL; Invitrogen, Ltd., Carlsbad, CA, USA) was added to each well, and the cells were stained for 1 h at room temperature. Finally, the cells were immersed in DAPI working solution (Boster Biological Technology, Ltd., Wuhan, China) for 10 min at room temperature. The stained cells were observed under a confocal microscope (Olympus FV3000; Olympus corporation; magnification, ×400). The quantification of co-localization of TβRI and CTxB was determined by ImageJ or Fiji analysis software. It was used to adjust the signal intensity thresholding and distinguish the region of interest (ROI). Then fluorescence co-localization (Mander’s correlation coefficient, MCC) was determined by evaluating fields acquired blindly in each group and by applying the same selection strategy [48,49,50]. The MCC obtained in each group was compared with the control group.

### 4.8. Statistical Analysis

GraphPad Prism 8.0 was used to statistically analyze the data. The Tukey post-hoc test was used to assess the differences between the two groups. One-way analysis of variance (ANOVA) was used to analyze the differences between more than two groups. All experiments were repeated three times. *p* < 0.05 indicated statistically significant differences.

## Figures and Tables

**Figure 1 metabolites-11-00562-f001:**
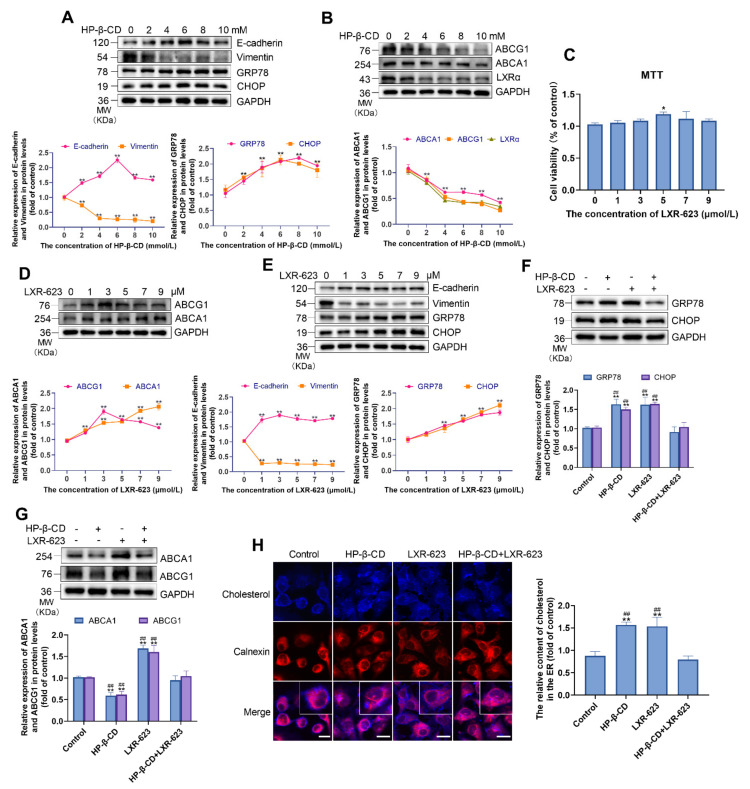
HP-β-CD caused the deposition of cholesterol in ER and elicited ER stress via decreasing the efflux of cholesterol in MDA-MB-231 cells, LXR-623 reversed this effect. (**A**) MDA-MB-231 cells were treated with various concentration of HP-β-CD (0, 2, 4, 6, 8, 10 mmol/L). Expression levels of E-cadherin, Vimentin, GRP78, and CHOP in MDA-MB-231 cells. (**B**) Expression levels of ABCG1, ABCA1, and LXRα in MDA-MB-231 cells. (**C**) MDA-MB-231 cells were treated with various concentration of LXR-623 (0, 1, 3, 5, 7, 9 μmol/L). Viability of MDA-MB-231 cells. (**D**) Expression levels of ABCG1, ABCA1 in MDA-MB-231 cells. (**E**) Expression levels of E-cadherin, Vimentin, GRP78, and CHOP in MDA-MB-231 cells. (**F**) MDA-MB-231 cells were treated with HP-β-CD then with or without LXR-623. Expression levels of CHOP and GRP78. (**G**) Expression levels of ABCA1 and ABCG1. (**H**) Co-staining of Filipin and immunofluorescence analyses for the content of cholesterol in ER. Magnification, ×400. The red fluorescence showed the ER. Scale bars represent 40 μm. The concentration of HP-β-CD and LXR-623 in (**G**,**H**) are 5 mmol/L and 3 μmol/L, respectively. Data are presented as the mean ± SD (*n* = 3). * *p* < 0.05 versus the control group, ** *p* < 0.01 versus the control group, ^##^
*p* < 0.01 versus the HP-β-CD + LXR-623 group.

**Figure 2 metabolites-11-00562-f002:**
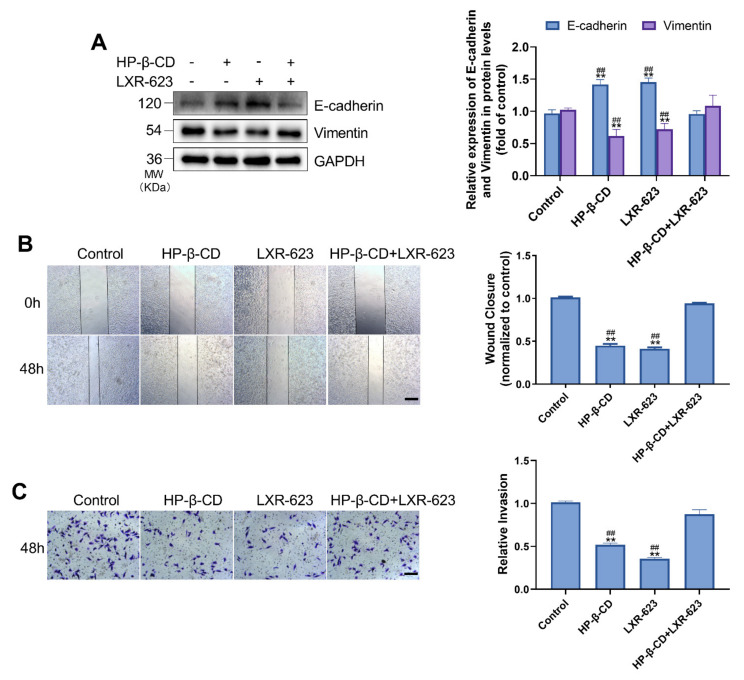
HP-β-CD inhibited the migration and invasion of MDA-MB-231 cells, LXR-623 reversed this effect. (**A**) MDA-MB-231 cells were treated with HP-β-CD then with or without LXR-623. Expression levels of E-cadherin and Vimentin. (**B**) Wound-healing assay and (**C**) Transwell invasion assay were used to evaluate the migratory and invasive ability of MDA-MB-231 cells. Data are presented as the mean ± SD (*n* = 3). Scale bars in (**B**,**C**) represent 250 μm. The concentration of HP-β-CD and LXR-623 in (**A**–**C**) are 5 mmol/L and 3 μmol/L, respectively. ** *p* < 0.01 versus the control group, ^##^
*p* < 0.01 versus the HP-β-CD + LXR-623 group.

**Figure 3 metabolites-11-00562-f003:**
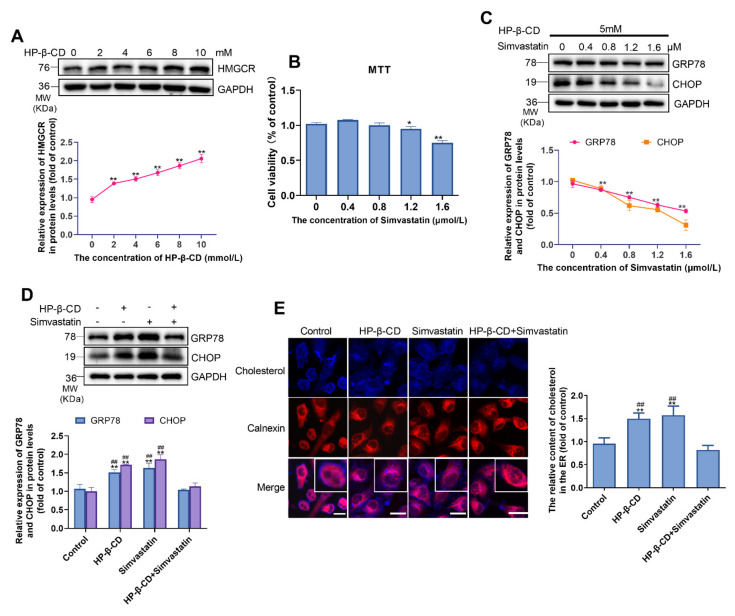
HP-β-CD caused the deposition of cholesterol in ER and elicited ER stress via increasing the synthesis of cholesterol in MDA-MB-231 cells, simvastatin reversed this effect. (**A**) Expression level of HMGCR. (**B**) MDA-MB-231 cells were treated with 5 mmol/L HP-β-CD with various concentration of simvastatin (0, 0.4, 0.8, 1.2, 1.6 μmol/L). Viability of MDA-MB-231 cells. (**C**) MDA-MB-231 cells were treated with 5 mmol/L HP-β-CD and various concentrations of simvastatin (0, 0.4, 0.8, 1.2, 1.6 μmol/L). Expression levels of GRP78 and CHOP. (**D**) MDA-MB-231 cells were treated with HP-β-CD then with or without simvastatin. Expression levels of GRP78 and CHOP. (**E**) Co-staining of Filipin and immunofluorescence analyses for the content of cholesterol in ER. The red fluorescence showed the ER. Scale bars represent 40 μm. The concentration of HP-β-CD and Simvastatin in (**D**,**E**) are 5 mmol/L and 1.2 μmol/L, respectively. Data are presented as the mean ± SD (*n* = 3). * *p* < 0.05, ** *p* < 0.01 versus the control group, ^##^
*p* < 0.01 versus the HP-β-CD + simvastatin group.

**Figure 4 metabolites-11-00562-f004:**
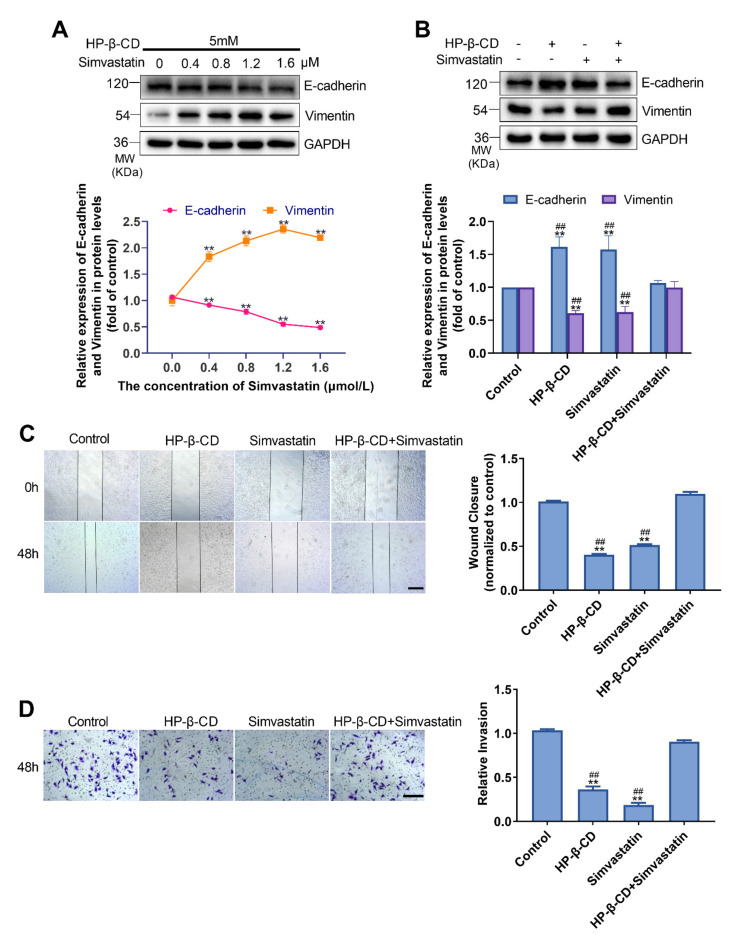
HP-β-CD inhibited the migration and invasion of MDA-MB-231 cells, simvastatin reversed this effect. (**A**) MDA-MB-231 cells were treated with 5 mmol/L HP-β-CD then with or without a various concentration of simvastatin (0, 0.4, 0.8, 1.2, 1.6 μmol/L). Expression levels of E-cadherin and Vimentin. (**B**) MDA-MB-231 cells were treated with HP-β-CD then with or without simvastatin. Expression levels of E-cadherin and Vimentin. (**C**) Wound healing assay and (**D**) Transwell invasion assay were used to evaluate the migratory and invasive ability of MDA-MB-231 cells. Scale bars in (**C**,**D**) represent 40 μm. The concentration of HP-β-CD and Simvastatin in (**B**,**D**) are 5 mmol/L and 1.2 μmol/L, respectively. Data are presented as the mean ± SD (*n* = 3). ** *p* < 0.01 versus the control group, ^##^
*p* < 0.01 versus the HP-β-CD + simvastatin group.

**Figure 5 metabolites-11-00562-f005:**
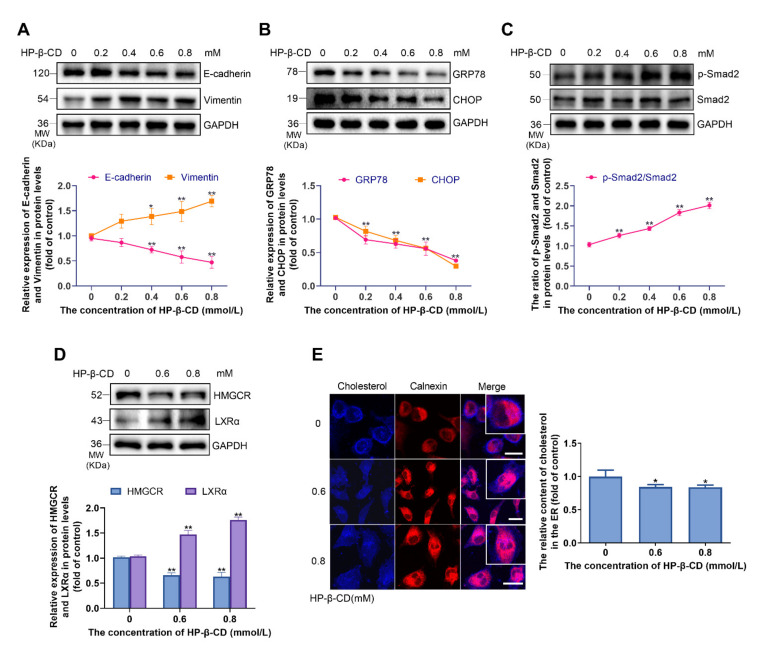
The effect of low concentrations of HP-β-CD on ER stress, TGF-β/Smad signaling pathway, and the distribution of cholesterol in MDA-MB-231 cells. (**A**) MDA-MB-231 cells were treated with various lower concentration of HP-β-CD (0, 0.2, 0.4, 0.6, 0.8 mmol/L). Expression levels of E-cadherin and Vimentin. (**B**) Expression levels of GRP78 and CHOP. (**C**) Expression level of p-Smad2 and Smad2. (**D**) MDA-MB-231 cells were treated with various lower concentration of HP-β-CD (0, 0.6, 0.8 mmol/L). Expression levels of HMGCR and LXRα. (**E**) Co-staining of Filipin and immunofluorescence analyses for the content of cholesterol in ER. Magnification, ×400. The red fluorescence showed the ER. Scale bars represent 40μm. Data are presented as the mean ± SD (*n* = 3). * *p* < 0.05, ** *p* < 0.01 versus the control group.

**Figure 6 metabolites-11-00562-f006:**
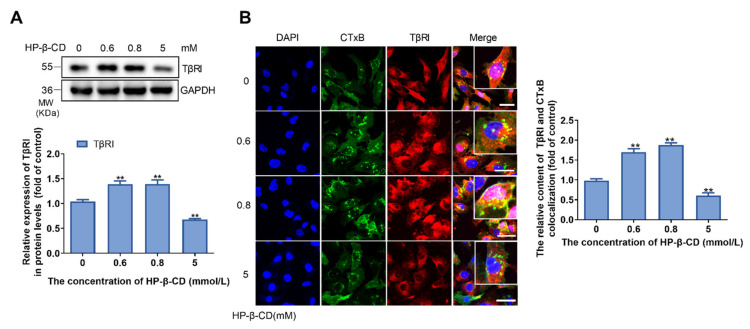
The effect of different concentrations of HP-β-CD on the expression and distribution of TβRI. (**A**) MDA-MB-231 cells were treated with various concentration of HP-β-CD (0, 0.6, 0.8, 5 mmol/L). Expression level of TβRI. (**B**) Confocal microscope image for immunofluorescence staining of TβRI and lipid raft. Magnification, ×400. Analyses for the content of CTxB and TβRI colocalization. The green fluorescence showed the lipid rafts, red fluorescence showed the TβRI. Scale bars represent 40 μm. Data are presented as the mean ± SD (*n* = 3). ** *p* < 0.01 versus the control group.

**Figure 7 metabolites-11-00562-f007:**
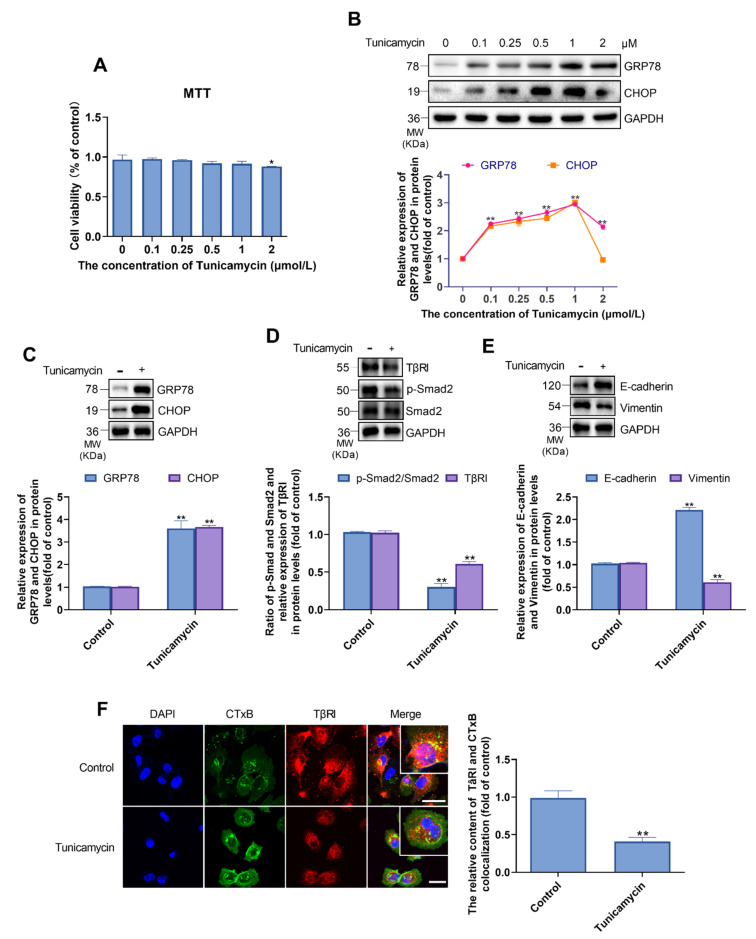
In MDA-MB-231 cells, tunicamycin-induced ER stress inhibited EMT via decreasing the content of TβRI in lipid rafts. (**A**) MDA-MB-231 cells were treated with various concentration of tunicamycin (0, 0.1, 0.25, 0.5, 1, 2 μmol/L). Viability of MDA-MB-231 cells. (**B**) Expression levels of GRP78 and CHOP. (**C**) MDA-MB-231 cells were treated with or without tunicamycin. Expression levels of GRP78 and CHOP. (**D**) Expression levels of TβRI, p-Smad2, and Smad2. (**E**) Expression levels of E-cadherin and Vimentin. (**F**) Confocal microscope image for immunofluorescence staining of TβRI and lipid rafts. Magnification, ×400. Analyses for the content of CTxB and TβRI colocalization. The green fluorescence showed the lipid rafts, red fluorescence showed the TβRI. Scale bars represent 40 μm. The concentration of tunicamycin in (**C**–**F**) is 1 μmol/L. Data are presented as the mean ± SD (*n* = 3). * *p* < 0.05, ** *p* < 0.01 versus the control group.

**Figure 8 metabolites-11-00562-f008:**
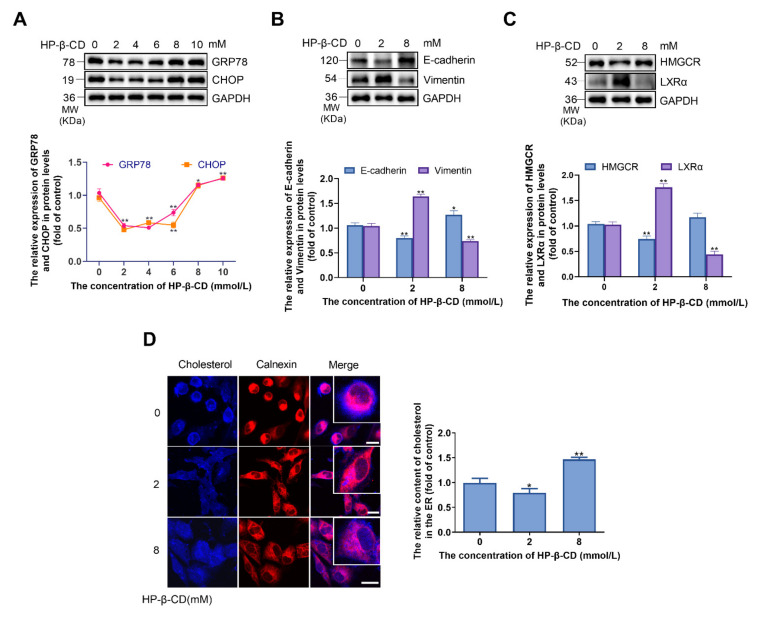
High concentration of HP-β-CD caused the deposition of cholesterol in ER and elicited ER stress via disturbing the synthesis and efflux of cholesterol in BT-549 cells. (**A**) BT-549 cells were treated with various concentration of HP-β-CD (0, 2, 4, 6, 8, 10 mmol/L). Expression levels of GRP78 and CHOP. (**B**) BT-549 cells were treated with various concentrations of HP-β-CD (0, 2, 8 mmol/L). Expression levels of E-cadherin and Vimentin. (**C**) Expression levels of HMGCR and LXRα. (**D**) Co-staining of Filipin and immunofluorescence analyses for the content of cholesterol in ER. Magnification, ×400. The red fluorescence showed the ER. Scale bars represent 40 μm. Data are presented as the mean ± SD (*n* = 3). * *p* < 0.05, ** *p* < 0.01 versus the control group.

## Data Availability

The data presented in this study are available in article.

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
