# Peer review of "2-Hydroxypropyl-β-cyclodextrin Regulates the Epithelial to Mesenchymal Transition in Breast Cancer Cells by Modulating Cholesterol Homeostasis and Endoplasmic Reticulum Stress"

_metabolites, 2021, doi:10.3390/metabo11080562_

Round 1
Reviewer 1 Report
Review Zhao et al. metabolites-1265510
Zhao and colleagues present a manuscript describing the effects of HP-beta-cyclodextrin (HP-b-CD) and LXR-23 on cholesterol levels and how these impact on ER stress and epithelial-mesenchymal transition in breast cancer cells. The study is in principle interesting, but not yet in a stage where publication can be recommended.
Specific points to be addressed are
-Page (p.) 4, line (l) 178 ff. the description of the results in Fig. 1A is too brief, it only contains the interpretation. First it should be described what is seen, then what this means. It is not obvious to everyone that increase in E-Cad and decrease in vimentin indicates EMT and likewise it is not clear that GRP78 and CHOP are ER stress markers.
“cholesterol efflux was found to be attenuated” is an overstatement, you did not measure cholesterol levels here. You can speculate that there is a reduced efflux via ABCA1 and ABCG1, but you have not shown this.
l. 185, explain LXR-623.
-Fig 1. Graphs: legend much too small, illegible for me.
-Fig. 1H, 3E, 5E, 6B, etc.: I´m not convinced of the stainings and the conclusions. The resolution is much too low and the image too small. Enlarged selected areas should be shown, but I doubt that you can unequivocally discern fillipin staining on the membrane from the ER staining and from other intracellular staining. The resolution of a confocal is maybe 300nm under optimal conditions, in z-direction much less, maybe optimistically around 500nm. How thick are your cells? 0.5-1 micrometer, maybe, impossible to make precise statements about localization, unless you see clear, typical structures of for example reticular network or typical plasma membrane localization where adjacent cells touch each other. No such structures are visible. Please also explain what the arrowhead shows and how the quantification was done, maybe with a supplemental figure where you clearly mark the structures that you define and quantify as ER or plasma membrane.
-Western Blots on all figures. It would be nice to see full-size membranes (or the stripes that you cut) as supplemental figure. It seems to me that the blots in the respective figures are not all from the same gel. If true, please indicate that in the legend and clearly state that the loading control can only be the loading control for some of the blots shown.
p. 5, l. 206 ff. Better write “HP-b-CD or LXR-623” then it is more clear that these are separate treatments.
p.6, l. 260. Overstatement, you did not measure cholesterol synthesis or uptake.
Fig. 7F. To me the pattern of the two stainings looks completely different. If there would be TbRI in rafts together with CTxB, you would expect at least some of the TbRI-staining in the dot-like CTxB-structures. Instead, the tbRI looks like an uniform, cytoplasmic protein, not like a plasma membrane receptor. Do you have a control for this antibody, staining of knock-out cells or similar? Also in this case it remains obscure how you generated the graph. Please show in detail how the quantification was made. What do the arrows indicate?
p.16. Discussion. Please remove the first three sentences. l.420ff the explanation about HP-b-CD better fit to the introduction.
l. 434, better in introduction
p.17, l 464. Show the data. How do they fit to your earlier statements that cholesterol efflux and uptake are affected?
Reviewer 2 Report
In this manuscript, the authors have reported that 2-hydroxypropyl-β-cyclodextrin (HP-β-CD) at low concentrations causes cholesterol depletion in the membrane, inhibits ER stress, and induces EMT through upregulation of TGF-β signaling. However, a higher concentration of HP-β-CD has a reciprocal effect.
While no major issues have been found with the manuscript, the authors are advised to resolve the following minor concerns:
The concentrations of HP-β-CD and LXR-623 used to treat the cells in Fig 1G-H and 2A-C have not been mentioned. Similarly, HP-β-CD and Simvastatin concentrations are missing from Fig 3D-E and 4B-D. Likewise, there is no mention of Tunicamycin concentrations in Fig 7C-F.
The authors should provide the concentrations used in the respective legends to the figures.
In line 377, replace ‘for the expression’ with ‘on the expression’.
Reviewer 3 Report
In this manuscript, authors investigated the role of HP-b-CD on ER stress, EMT and cholesterol homeostasis. Authors conducted various experiments and showed interesting data. I have only few comment related editing error.
L238-239
It is not easy to understand the sentence so please rewrite the sentence.
L415-418 (the first paragraph of the discussion section)
I think there are several “instruction (how to describe the discussion section) for authors” sentences mistakenly inserted.
Round 2
Reviewer 1 Report
Review Zhao et al. metabolites-1265510 V2
Zhao and colleagues present a revised manuscript describing the effects of HP-beta-cyclodextrin (HP-b-CD) and LXR-23 on cholesterol levels and how these impact on ER stress and epithelial-mesenchymal transition in breast cancer cells. The authors made a great effort and mostly addressed the points I raised. Before final acceptance the following small issues should be dealt with:
Specific points to be addressed are
- My previous comment on cholesterol efflux
“cholesterol efflux was found to be attenuated” is an overstatement, you did not measure cholesterol levels here. You can speculate that there is a reduced efflux via ABCA1 and ABCG1, but you have not shown this.
Your new version
After the treatment of HP-β-CD, there is a reduced cholesterol efflux via a decrease in the expression of LXRα, ABCA1, and ABCG1 (Fig. 1B).
You rephrased the sentence without changing the meaning. But since you haven´t measured efflux, you should write: After the treatment of HP-β-CD we observed a decrease in the expression of LXRα, ABCA1, and ABCG1, suggesting a reduced cholesterol efflux.
- Fig. 7f
I´m still not convinced that what you see is actually TβRI, but let´s just leave it at that. I would never trust images from the vendor of the antibody. I´m sure you, like anyone else in cell biology, spent thousands of renminbi in non-functional antibodies….
But just to clarify: TβRI is certainly not in the cytoplasm, it is a membrane protein, as you (partially) correctly illustrated in the graphical abstract. If it is not in the plasma membrane, it can only be localized in intracellular compartments of the secretory pathway, for example the ER or the Golgi or secretory vesicles, but never just in the plasma membrane. Such a staining would look different from a typical, “real” cytoplasmic staining. Actually your stainings, as far as one can tell, don´t look like a cytoplasmic staining.
Please revise the left part of the graphical abstract and redraw the cytoplasmic TβRI in a vesicle to visualize it is embedded in a membrane and not in the cytoplasm.
